# "I Had No Idea about This:" A Mixed-Methods Exploration of Sexual Health and HIV Prevention Needs among Black Youth in a Southern City

Allysha C. Maragh-Bass [1,2,*], John T. Mitchell [3], Marie C. D. Stoner [4], Nivedita L. Bhushan [4], Linda Riggins [5], Alexandra Lightfoot [5] and Amy Corneli [6]

1 Global Health and Population Division, FHI 360, Durham, NC 27701, USA
2 Duke Global Health Institute, Durham, NC 27710, USA
3 Department of Psychiatry and Behavioral Sciences, Duke University Medical Center, Durham, NC 27710, USA; john.mitchell@duke.edu
4 Women's Global Health Imperative, RTI International, Berkeley, CA 94704, USA; mcstoner@rti.org (M.C.D.S.); nbhushan@rti.org (N.L.B.)
5 Department of Health Behavior, University of North Carolina Gillings School of Public Health, Chapel Hill, NC 27599, USA; lriggins001@gmail.com (L.R.)
6 Department of Population Health Sciences, Duke University, Durham, NC 27708, USA; amy.corneli@duke.edu
* Correspondence: amaraghbass@fhi360.org; Tel.: +1-919-744-5040 (ext. 11966)

**Abstract:** HIV disparities continue to persist among Black youth in the South. We conducted quantitative surveys (N = 83) and follow-up qualitative interviews (n = 13) to assess sexual health needs including but not limited to Pre-Exposure Prophylaxis (PrEP) awareness. Participants all identified as Black; most survey respondents identified as being assigned female sex at birth and heterosexual. Both qualitatively and quantitatively, knowledge about HIV and PrEP and engagement in sexual health preventive behaviors was low. Participants described a need for more comprehensive sexual health education at younger ages and more routinized HIV testing. The latter was recommended even for people with a low perceived risk of HIV. Future studies should explore how to tailor communication to youth of color specifically and understand nuances of experiences they may have based on their sexual orientation and gender identity to promote engagement in sexual health preventive care, including but not limited to HIV prevention and PrEP uptake. The findings suggest that comprehensive sexual care that provides education on HIV, STI, and pregnancy prevention is critical for youth experiencing inequities in a Southern city context.

**Keywords:** adolescent health; African Americans; health equity; HIV prevention; public health; social determinants of health

## 1. Introduction

Adolescents 13 to 24 years old account for 22% of new HIV infections in the US, with 55% of these cases occurring among Black adolescents, particularly among Black sexual and gender minority (SGM) youth [1]. Despite making up 37% of the US population, about 45% of people with an HIV diagnosis live in the Southern US [1]. Adolescents and young adults in the US South bear some of the highest rates of HIV in the country [1,2]. Pre-Exposure Prophylaxis (PrEP) is an efficacious biomedical intervention to prevent HIV, which was approved for use among adults in 2012 and among adolescents in 2018 [3,4]. Despite PrEP's efficacy, awareness is variable among those who may potentially benefit from it and is particularly low among adolescents and young adults. Studies that include adolescents indicate that PrEP awareness ranges from 11 to 74% [5–12]. These rates vary as a function of different factors but overall indicate wide variability in PrEP awareness. In addition, perceived discrimination against those with HIV were each associated with decreased PrEP awareness in a sample of men who have sex with men (MSM) who identified as Black in

previous research [5]. In another study of MSM, those who were younger were less likely to have heard about PrEP, and men identifying as Black were more likely to state they would not use PrEP because of side effect concerns [9]. These findings highlight the need to ensure that SGM youth—particularly youth and young adults of color—are aware of PrEP and that it is accessible.

Studies evaluating the feasibility and acceptability of PrEP in adolescents have indicated that even when enthusiastically taken up, adolescents and young adults have challenges with daily dosing fatigue, perceptions of stigma, and worries about side effects that may impact their adherence [13–15]. Further, access and uptake of PrEP pose challenges outside of the clinical trial setting, as young people are the least likely of any age group to have access to HIV prevention services and to be linked to care [1,2]. Problems with the uptake and sustained use of PrEP may be due to inadequate comprehensive sexual health education or knowledge of PrEP, perceived lack of risk, stigma from parents or healthcare workers, or lack of social and partner support. Black SGM youth who have a disproportionately high likelihood of HIV exposure also face discrimination based on race, sexual orientation, and/or gender identity [5,7,8,10,11].

Understanding awareness and attitudes toward PrEP among Black adolescents is important to improve dissemination and PrEP uptake. We explored PrEP awareness and attitudes among Black adolescents and young adults in a Southern city using a mixed-methods approach. We enrolled adolescents and young adults (aged 13–24) to complete a quantitative survey on topics related to PrEP and HIV risk, followed by a subsample that participated in an in-depth qualitative interview. The objective of this study was to characterize broader sexual health needs and PrEP attitudes among Black adolescents and young adults in the South. Informed by our prior research, which suggested that youth are not engaged in healthcare overall [16], we also qualitatively explored barriers to care engagement for sexual health more broadly and not only specific to HIV and PrEP. We adopted a mixed-methods approach: qualitative methods were used to examine the awareness and attitudes towards PrEP among Black adolescents, which were further characterized with quantitative findings [17–19].

## 2. Methods

### 2.1. Recruitment and Eligibility

We conducted a quantitative survey followed by in-depth qualitative interviews in a Southern city. Our county has the fourth highest reported HIV infections in our state, with 25.7 cases per 100,000 (versus the state rate of 19.3); the high HIV prevalence along with existing previously identified HIV prevention needs from our prior research were why we selected this city [1,2,16]. We also note that our city is only 90 min from the city with the highest rates of HIV in the state; many of our young adults move constantly between these studies and are navigating rates of HIV that are nearly as high as rates in metropolitan hubs such as Miami and Atlanta [1,2,16]. Eligibility criteria for the survey study included being aged 13–24 and self-identifying as Black. Survey participants were recruited through community partners and listservs in our city, a housing program, a PrEP clinic, local LGBTQ advocacy groups, and other community organizations working with young people. To ensure that the perspectives of youth, those on PrEP, and those who identify as SGM were included, we also recruited through organizations working with youth and those that prescribed PrEP or provided PrEP referrals [16]. This study received Institutional Review Board (IRB) approval including IRB reliance agreements across our respective institutions. Participants were asked to consent to completing screener questions, and the survey required verbal consent; individuals who were interested in being contacted for additional study activities were invited to complete interviews which required written consent (described further below).

### 2.2. Study Procedures

Quantitative data collection: For the online survey, respondents were asked to complete a brief PrEP awareness and attitudes survey. The survey questions in total ranged from 30 to 50 questions based on skip patterns and how respondents answered; they took on average 18 min to complete. The survey was administered over REDCap, a secure, web-based application designed to support data capture. Eligible participants were sent the link via email and completed the survey online remotely. To encourage participants to respond openly to these questions, we informed participants that names would not be linked with their survey responses. Written consent was also waived for the survey so we could have one fewer document with participants' names; as a result, minor assent was also not required since participation was confidential, surveys could be discontinued, and further study participation was optional.

If survey participants were interested and willing to be contacted for additional study activities, they were invited to give their contact information (name, email, phone) for our team to follow up directly. These individuals were our pool of qualitative participants and were purposively selected by our team for interviews based on factors such as age, sex at birth, and gender identity. At the start of the interviews, team members verbally explained the consent form (emailed to participants in advance) and gave them an opportunity to ask questions. Team members then signed consent forms indicating they received verbal consent from participants prior to initiation of the interview.

Qualitative data collection: We conducted qualitative in-depth interviews with a subsample of youth who participated in the survey to gain a deeper understanding of PrEP awareness and attitudes. The interview involved a guided discussion with interviewer prompts about PrEP awareness and attitudes; interviews were conducted by a study team member via web-conference on Zoom (Version 5.0) and lasted between 45 and 70 min. Questions were omitted if clients were not comfortable answering them and/or if they were not relevant to clients (e.g., if they were not on PrEP). Interviewers kept detailed notes and memos of experiences seeking sexual health preventive care to note differing observations by assigned sex at birth, gender identity, sexual orientation, and age (specifically younger versus older than 21). Interviews were audio-recorded and transcribed verbatim.

### 2.3. Measures

*Quantitative:* PrEP awareness was assessed with the question "Have you heard of the once-daily pill to prevent HIV called Pre-Exposure Prophylaxis (PrEP)?" If participants had heard about PrEP, they were asked questions about where and if they had taken it. The survey also assessed basic demographics, sexual orientation, history of HIV testing, perceived HIV risk (lifetime), and history of sexually transmitted infections.

*Qualitative:* In-depth interview questions were related to perceived HIV risk, engagement in HIV risk behaviors, history of HIV prevention service utilization, PrEP stigma and perceptions (both as it relates to themselves and others), structural barriers to healthcare, history of racial and sexual discrimination, stigma, and HIV and HIV risk behaviors. While our surveys were conducted in late 2019, our interviews we still ongoing in early 2020. Therefore, we included questions exploring participant experiences since the onset of the COVID-19 global pandemic (see Table 1). For participants who had ever been prescribed PrEP, we also asked questions about additional factors that facilitated use and barriers to retention in care and perceived changes in HIV-preventive behaviors.

**Table 1.** Interview domains and sample prompts.

| **Overview and Introduction** |
| --- |
| • Can you briefly walk me through what a typical day is like for you? |
| • What are some good things about how things are going in your life? What about the tough things going on? Since COVID-19? |

| **General Stressors** |
| --- |
| • What are the main things stressing you right now? |
| • What happens to you when you usually feel stressed *[prompt: mentally, physically, or affecting others]*? |
| • How are you dealing with your stress right now? What has helped? What else might help? |

| **HIV-Related Experiences and Perceptions (less specific to PrEP)** |
| --- |
| • Have you ever been to a clinic to get sexual health care? |
| • When you think of people with HIV, what comes to mind? |
| • How do you currently protect yourself from getting HIV *[prompt: PrEP, condoms, testing, abstinence]* |
| • How do you think stigma affects sexual health? How does it affect the Durham community? |
| • How do you think adolescents and young people in Durham could be better supported to fight this stigma? Discrimination? |

| **PrEP Awareness and Experience** |
| --- |
| • What have you ever heard about PrEP? |
| • So what do you think about PrEP? Is it a good thing or a bad thing? What makes you feel that way? |
| • Have you ever taken Truvada or Descovy *[or say PrEP]*? |
| • What is the best thing we can do to support young people who want to take PrEP? |

| **Barriers to HIV Prevention (not already discussed)** |
| --- |
| • How easy is it to go see a healthcare provider, like a doctor or a nurse? |
| • How comfortable are you talking about sex with a healthcare provider? About HIV? |
| • Who is a trusted adult you talk to about sex and/or HIV? How do they support you? What other support do you need from them? |
| • What has been going on in the community that has affected your health? Going to a healthcare provider? How so? |

| **Conclusion** |
| --- |
| • Before we wrap up, what should people know about your sexual health needs? HIV information needs? PrEP information needs? |
| • What topics should we ask about, perhaps, in future interviews with other young adults in our project? |
| • What else would you like to mention that matters to you? |

## 2.4. Data Analysis

Quantitative survey responses about PrEP attitudes and experiences were examined descriptively overall and by gender and age (age 13–19 and 20–24)—we determined a priori that no inferential statistics would be calculated due to the exploratory focus of our study and small sample size. We report numbers and percentages for categorical variables and medians for the interquartile range for continuous variables.

Qualitative analyses were conducted using NVivo Version 12.0 [20]. Our deductive analytical approach allowed for excerpts of transcribed interviews to be identified and associated with different categories from the interviews (e.g., comments pertaining to PrEP barriers). Similar to other qualitative studies [21–24], the investigative team used a multi-step rapid qualitative thematic analysis process using the following 5 steps: (1) all transcripts were read multiple times for understanding prior to coding; (2) structural coding was used to group corresponding excerpts in broad interview categories which were then reviewed to determine information saturation [saturation determined by interviewer consensus]); (3) categories with saturated data were then reviewed to identify whether multiple categories within each broad category were needed (i.e., if data needed to be double-coded to accurately depict what participants discussed); (4) all interviewer analytic memos (summarized by the interviewer after each interview) were reviewed multiple times; and (5) information-saturated categories were finalized along with exemplar quotes after multiple rounds of discussion with the principal investigators [21–24]. Coding and other analytical steps were conducted by one of the principal reviewers (AMB), who is a graduate-trained mixed methodologist, and findings were reviewed by another principal investigator (JM) and a third study team member, who conducted the qualitative data collection and is also a graduate-trained mixed methodologist (NB) [25].

Although during data collection, we conducted the survey first followed by the qualitative interviews, we used mixed-methods integration of findings with a QUAL→quant approach during analysis [26]. The qualitative analysis was conducted first, in order to identify the most salient information from the interviews, which was then matched to corresponding survey questions. We used this approach to allow for the quantitative findings to elaborate upon the qualitative findings and triangulate the findings to help best characterize the experiences of the participants.

*Sex- and gender-based analyses.* In our quantitative survey, we asked participants their 'sex at birth' as well as their 'current gender identity.' Sex at birth had male/female binary responses, while current gender identity had multiple categorical options including 'transgender woman, transgender man, nonbinary, genderqueer, and other (with free response option).' We made these necessary distinctions given our inclusion criteria were not specific to either sex or gender-based recruitment quotas. Qualitative analyses provided necessary context for quotes and were tagged with demographic data such as the participant's current gender identity and age.

### 3. Results

#### 3.1. Demographic Characteristics

A total of 83 adolescents and young adults (aged 13–24) completed the quantitative survey, and 13 participated in the follow-up qualitative in-depth interviews. Participant demographics are summarized in Table 2 (survey study) and Table 3 (interview study). As shown in Table 3, all survey participants were Black and aged 13–24 years (N = 83). Roughly two-thirds of survey respondents were aged 18 or older (68%; n = 54), and most were assigned female sex at birth (75%; n = 62). Nearly 30% had (1) experienced food insecurity in the prior six months (28%); and (2) recalled a time they needed sexual health services but did not receive them (28%; n = 23). Over 60% of respondents had never been tested for HIV (63%; Table 3).

**Table 2.** Survey participant characteristics, N = 83.

| Demographic Characteristics | N (%) |
|---|---|
| **Race** | |
| Black | 83 (100%) |
| **Age [a]** | |
| 13–17 | 26 (32.5%) |
| 18–20 | 25 (31.3%) |
| 21–24 | 29 (36.3%) |
| **Gender** | |
| Male | 21 (25.3%) |
| Female | 57 (68.7%) |
| Transgender | 5 (6.0%) |
| **Sex Assigned at Birth** | |
| Male | 21 (25.3%) |
| Female | 62 (74.7%) |
| **Sexual orientation** | |
| Homosexual | 10 (12.0%) |
| Heterosexual | 47 (56.6%) |
| Bisexual | 13 (15.6%) |
| Queer | 2 (2.4%) |
| Not sure | 1 (1.2%) |
| Other | 8 (9.6%) |
| Decline to respond | 2 (2.4%) |
| **Food insecurity (past 6 months)** | |
| No | 59 (71.9%) |
| Yes | 23 (28.1%) |
| **Employment status [a]** | |
| Unemployed | 39 (46.9%) |
| Part-time employment | 24 (28.9%) |
| Full-time employment | 14 (16.9%) |
| **Preferred health information resource [a]** | |
| Parent | 28 (33.7%) |
| Friend | 2 (2.4%) |
| Sibling/family member | 1 (1.2%) |
| Internet | 20 (24.1%) |
| Healthcare provider | 31 (37.4%) |
| **Have needed sexual health services and did not receive them** | |
| No | 60 (72.3%) |
| Yes | 23 (27.7%) |
| **Ever received an HIV test** | |
| No | 52 (62.7%) |
| Yes | 31 (37.4%) |

**Table 2.** *Cont.*

| Demographic Characteristics | N (%) |
|---|---|
| **Lifetime perceived HIV diagnosis likelihood** | |
| Not at all likely | 44 (53.0%) |
| A little likely | 26 (31.3%) |
| Somewhat likely | 9 (10.8%) |
| Very likely | 4 (4.8%) |
| **PrEP awareness prior to study enrollment** | |
| No | 37 (44.6%) |
| Yes | 46 (55.4%) |
| **If aware of PrEP, where learned of PrEP** | |
| A friend or a sex partner | 4 (8.7%) |
| An advertisement/commercial | 27 (58.7%) |
| Student health clinic | 2 (3.3%) |
| A student organization at school | 3 (6.5%) |
| Social media | 2 (3.3%) |
| Other | 8 (17.4%) |
| Decline to answer | 0 (0%) |
| **PrEP taken previously** | |
| Yes | 3 (3.7%) |
| No | 80 (96.3%) |
| **PrEP taken currently** | |
| Yes | 1 (1.2%) |
| No | 82 (98.8%) |
| **Willing to tell others I'm on PrEP** | |
| Yes, only sexual partners | 23 (27.7%) |
| Yes, close friends or sexual partners | 14 (16.9%) |
| Yes, anyone who asks | 27 (32.5%) |
| Yes, I'd spread the word | 15 (18.1) |
| No | 4 (4.8%) |
| **If people knew I took PrEP, they'd see me** | |
| Negatively | 26 (31.3%) |
| The same | 5 (6.0%) |
| Positively | 52 (62.7%) |

[a] Missing data, values do not sum to 100%.

**Table 3.** Interview participant demographics, N = 13.

| Demographic Characteristic | N(%) |
|---|---|
| **Age (years)** [a] | |
| 13–17 | 5 (38%) |
| 18–22 | 8 (62%) |
| **Race** | |
| Black | 13 (100%) |
| **Sex Assigned at birth** | |
| Male | 2 (15%) |
| Female | 11 (85%) |
| **Gender** [b] | |
| Male | 2 (15%) |
| Female | 8 (62%) |
| Transgender | 2 (15%) |
| Genderqueer/non-binary | 1 (8%) |
| **Sexual orientation** [c] | |
| Heterosexual | 9 (69%) |
| Bisexual | 1 (8%) |
| Queer | 1 (8%) |
| Other [d] | 2 (15%) |
| **Current grade in school** [e] | |
| 6th–8th grade | 1 (8%) |
| 9th–12th grade | 5 (39%) |
| Community college | 2 (15%) |
| Four-year university | 3 (23%) |
| Graduate school | 1 (8%) |
| Not in school | 1 (8%) |
| **Employment** | |
| Full-time | 2 (15%) |
| Part-time | 1 (8%) |
| Unemployed | 9 (69%) |
| Other (write-in option) [f] | 1 (8%) |

[a] Mean age was 18.15 (standard deviation = 2.76). The frequency for each age was 1 (8%) for 13 years old, 0 (0%) for 14 years old, 2 (15%) for 15 years old, 0 (0%) for 16 years old, 2 (15%) for 17 years old, 2 (15%) for 18 years old, 2 (15%) for 19 years old, 1 (8%) for 20 years old, 1 (8%) for 21 years old, and 2 (15%) for 22 years old. [b] "Two-spirited" was a response option, though it is not listed in the table because it was not selected. [c] "Homosexual", "Not sure", and "Decline to respond" were response options, though they are not listed in the table because they were not selected. [d] write-in responses included "pansexual" (n = 1) and "just me" (n = 1). [e] Highest grade completed for the one participant not currently in school was some college. [f] Among the write-in options, the one participant who used this wrote in "undetermined".

### 3.2. Qualitative Categories and Quantitative Findings

We present our qualitative interview findings in five categories, together with exemplar quotes and relevant descriptive statistics from the survey in Table 4.

**Table 4.** Qualitative themes and exemplar quotes along with descriptive frequencies from survey data.

| Qualitative Theme | Exemplar Quote | Relevant Quantitative Descriptive Results |
|---|---|---|
| *Impact of COVID-19 on daily life experiences*<br><br>*Participants described daily stress of early life during lockdown, such as mental health challenges and/or financial hardship* | So we lost our car and so I haven't really been going out that much because I really don't understand the transit system in—… before COVID I was seeing my niece a lot more because she has special needs and I love seeing her…that—was really, really hard for me … (African American/Nonbinary/18 years old)<br><br>Just the fact that COVID is around, that's a stress … keeping the mask on you at all times and just having to be clean, when I come home to my baby, no viruses, no infections, nothing comes home with me… For me, it affects me mentally because I get irritated, and then I just shut down, don't want to talk to nobody, just want to be alone. (African American/Female/20 years old) | Among our quantitative sample (N = 83), nearly 50% were unemployed. Among those who were young adults aged 18 and up (e.g., of definite working age), unemployment rates ranged from 30% to 44% depending on age category.<br><br>Similarly, nearly 30% overall reported experiences of food insecurity; the rates were marginally significantly higher among young adults aged 21 and up (41%; $p < 0.08$) |
| *Health information seeking, social interactions*<br><br>*Participants described what sexual health information they have sought, sources they consult, and social interactions they have had in their personal lives around sexual health* | [if I had questions] Probably my primary care doctor and then also people that look like me and share the same identities as me and ask about their experiences with using PrEP. (African American/Nonbinary/18 years old)<br><br>I definitely do [have trusted sources of sexual health information]. I don't really talk to my family about it…But I do talk to my girlfriend about it… my close friends about my experience…sometimes I'll ask my clinician, "Oh, can you tell me more about genital warts," because that's something that I have. But I've yet to actually go to my health care providers…based off of something like that … (African American/Transgender/19 years old)<br><br>… I don't think they [my family] would be mad [if I asked about this] but I think they would feel some type of way about it… they'll probably think, "Why is she asking this type of thing?" (African American/Female/15 years old) | Across all survey respondents, roughly one-third preferred their parents or guardian as a resource for health information (33.7%).<br><br>Nearly 40% preferred a healthcare provider (37.4%); roughly one-fourth preferred the internet (24.1%). |
| *Barriers to seeking sexual healthcare*<br><br>*Participants described cost and fear of awkward interactions with providers as barriers to seeking sexual healthcare services* | …So I go to—because I am trans, I get testosterone from there … the last time I went, they asked if I wanted to pee in a cup to check for any…I have access to it. But I also sometimes I'm not open to getting it, just because insurance won't always cover my me going to—(African American/Transgender/19 years old)<br><br>… [It would help to have providers] especially when they can relate to what I'm going through…there's been times where I've had doctors, they really understood my problems, and really understood what it's like to be in my shoes… It makes me want to be more honest with them. (African American/Male/18 years old) | Nearly 30% of respondents reported a time they needed sexual health services and did not receive them. Of these individuals, reasons included the following:<br>• Cost (22.7%);<br>• Embarrassment (31.8%);<br>• Uncertainty of where to go (13.6%);<br>• Fear of stigma and others finding out (9.1%);<br>• Too busy (9.1%). |

**Table 4.** *Cont.*

| Qualitative Theme | Exemplar Quote | Relevant Quantitative Descriptive Results |
|---|---|---|
| *HIV knowledge, PrEP familiarity*<br><br>*Participants described their knowledge of HIV, PrEP, outstanding questions, and sexual health needs around them* | **Perceptions of HIV:** . . . with me having a gay brother, I really get concerned about him and I make sure that. . . "Have you been tested for HIV". It's uncomfortable to ask that because I feel like he should already [be doing it] . . . (African American/Female/21 years old)<br><br>People who didn't take the right precaution to prevent it, I just feel maybe they weren't careful enough. But yeah, I just really am a strong advocate for people getting tested early because one thing I do know is that when you get tested earlier. . . people can stop being afraid and actually go get tested to protect their health and stuff . . . There's different ways it can be spread . . . (African American/Female/18 years old)<br><br>**Preventing HIV:**. . . the people I've been with, I know them and they haven't been with that many people or they haven't been with people at all. . . I can see if I was active every week or twice a week I would, but I've only been with two other people, so I don't feel like it's that big of a deal to me [to be tested]. . .(African American/Male/18 years old)<br><br>**PrEP:** I already said this but definitely pushing the PrEP pill because I had no clue what you guys were talking about at first. I was like, "What's this?" I did not know exactly what it is. I thought it was a two or three year thing, nah, [PrEP has been around] ten years?. . . For someone my age not knowing about it is kind of crazy. . . I feel like a lot of the questions I can have, I'll do research..(African American/Male/18 years old) | Over half of young adults aged 18–20 reported never receiving an HIV test. Sixty percent of young adults aged 21+ had never had an HIV test.<br><br>Only half of males and half of females overall knew about PrEP. Of those sexually active overall, only 51.7% had been tested for HIV. |

**Table 4.** *Cont.*

| Qualitative Theme | Exemplar Quote | Relevant Quantitative Descriptive Results |
|---|---|---|
| *Multilevel stigma and stigma reduction*<br><br>*Participants described multiple dimensions of stigma, and gave recommendations for addressing some forms* | **Transphobia/Queer discrimination**:. . .this [my high] school is kind of not being fair to trans folks and kind of dealing with that. But that was kind of a thing for a lot of my high school years, and then switching to another high school having to deal with that same battle again. (African American/Nonbinary/18 years old)<br><br>. . .I originally was very open to it more so when it came to health care providers. But after those instances, and even some of my family members who have just misgendered me on purpose, or just didn't believe me, like those instances have caused me to be close off health care providers that I know aren't going to support me. . .(African American/Transgender/19 years old)<br><br>**Community perceptions (HIV stigma)**. . .Depends on who you talk to you. Everybody has their own opinion. . .Negative ones I heard was people saying, "Oh, they're dirty, that they got HIV and stuff". (African American/Male/15 years old)<br><br>Stigma comes from because people aren't educated about it. And since you're not educated it's probably rare that you'll go get tested as well. And so, that means your transferring more stuff because you didn't know you was carrying it. . .. (African American/Female/22 years old) | Over 60% of survey respondents reported being viewed positively if people know they were taking PrEP, yet nearly one-third reported being viewed negatively (31.3%).<br><br>When asked who they were willing to tell that they were on PrEP, responses varied by who would be told (though most reported willingness to do so):<br>• Only sexual partners (27.7%);<br>• Only close friends and sexual partners (16.9%);<br>• Anyone who asked me (32.5%);<br>• I'd spread the word (18.1%);<br>• No (4.8%).<br>When asked about lifetime perceived risk of HIV, 65% of those aged 21+ reported their risk as the lowest. This same group were also sexually active, and most had never been tested for HIV. |

### 3.3. The Impact of COVID-19 on Daily Life Experiences and Sexual Health Needs

Qualitatively, participants described their experiences early in the COVID-19 global pandemic. Much of this included experiencing mental health challenges and stress related to the abrupt transition to lockdown, and social isolation as a result. For example, one participant noted the stress from different social interactions caused by COVID-19:

> *Okay, so during COVID I was I guess more interactive because I had to show up for school and stuff. . . And then COVID hit. . .I'm inside the house most of the time anyway. But yeah COVID has been stressing me out. Not necessarily because of the amount of deaths . . .(Black/Female/17 years old)*

Other participants described the stress of early lockdown mainly in the form of financial hardships like job loss. Some of them mentioned this impacting both their mental health and their ability to afford mental healthcare. They described losing their jobs as a stressful change and spending most of their time in isolation at home.

> *. . .Although I did lose my job a couple of months ago, because of COVID. And it was in the service industry. . .it did mess up my job and my ability to work [and afford therapy]. But other than that, it's just been me at home most of the time. . . my family is still safe, which is really good. . .(Black/Transgender/19 years old)*

A handful of youth also described 'not thinking about HIV or sex' due to limited social engagement and less concern with sexual encounters overall. However, this did not necessarily reflect a lack of sexual activity or less need for sexual health information. Quantitatively, nearly half of the survey sample was unemployed just before the COVID-19 global pandemic (when surveys were conducted). Among those 18 years old or older, unemployment was 44% among 18- to 20-year-olds and 28% among young adults aged 21 years and older. Nearly 30% (n = 24) of the overall sample reported that they had been concerned about having enough for themselves and their family in the past 6 months, particularly among participants ages 21 years and up.

### 3.4. Health Information Seeking and Social Interactions

Qualitatively, participants described multiple instances in which they had sought sexual health information, and social interactions they had related to the information they sought. Some participants shared that this was part of their daily social life conversations, such as with friends and loved ones, while others talked about seeking health information from healthcare providers to whom they had access. For example, one participant described not trusting school counselors and had a preference for using mobile apps for information related to sexual health.

> *. . .I guess the counselors are there and stuff but I don't honestly trust them. . .I could talk to [my mom], but I'm not sure what her reaction would be. . .one thing I do know is there is this app for women called—that helps us track our menstrual cycles and stuff and they have articles by very high-class healthcare professionals. . .(Black/Female/18 years old)*

Several participants described that beyond mistrust, they might experience stigmatization if they spoke to adults in their lives about sexual health. One participant described feeling as though she "had no business" asking questions about sexual health and felt that her family members would similarly wonder why she was inquiring about it. Another participant described preferred to seek sexual health information from people in their in their daily life because these individuals "look like me". Quantitatively, survey respondents varied widely in their preferred resources for seeking health information. Just over 30% preferred to speak to a parent (n = 28) and nearly 40% preferred speaking to a healthcare provider (37%; n = 30). Minors under the age of 18 preferred speaking to parents while adults ages 21 and up preferred speaking to healthcare providers.

*3.5. Barriers to Seeking Sexual Healthcare*

Qualitatively, participants described multiple types of barriers to seeking sexual healthcare services. Structural and socioeconomic barriers like cost and insurance were described most often. One participant described the ways in which they avoid paying for sexual healthcare that they need, such as sending electronic messages to providers.

> *It's hey, can I afford this doctor appointment? Can I afford the medicine that I'll need from this appointment? And just sending a message on my chart, first to talk to my doctor about it, would be the best bet and just doing my own self research. And if there's something I can do at home, I'd probably seek this out first before I go to the doctor. . .it's just not as accessible as it used to be for me. (African American/Nonbinary/18 years old)*

Another described that although they have access to sexual health information and have special needs for service provision because they are transgender and need hormone therapy, they do not access those services, "because insurance won't always cover me going".

Participants also described discomfort and fear of awkward interactions as a deterrent from seeking care. Participants explained that these feelings were different from the perception of being stigmatized. Several described the awkwardness due to being young and having their parents involved and feeling uncomfortable even when providers say "no, your parents don't have to know". Several described that even when the parent or guardian was not in the room, it was still awkward to talk to providers about sex. A participant stated that one way this could be improved would be for providers to approach the conversation by giving sexual health information first and then asking questions to follow up for clarification.

> *. . .maybe it should be a different approach to where they ask, "Are you sexually active,". . .Then, sometimes they do it in front of your parents and you're not going to get the correct answer most of the times if you're doing it in front of a parent [so time alone with a provider could be good]. . . start by giving information about it before they actually ask that question [to make it less awkward]. . . (Black/Female/22 years old)*

Quantitatively, roughly 28% (n = 23) of survey respondents reported a time they needed sexual health services and did not receive them. Of those 23 respondents, seven did not seek services because of embarrassment, while two reported a fear of stigma. Nearly one-fourth (n = 5) described cost as the reason for not seeking sexual health services.

*3.6. HIV Knowledge and PrEP Awareness*

Qualitatively, participants described HIV as something that they perceive to "usually affect gay people". Others described that their first impression of HIV is knowing that people "did not take the right precaution" to prevent it. When asked about preventing HIV, for example, one participant who was sexually active described not needing to get tested for HIV because she is young and "I feel like the people I've been with, I know them and they haven't been with that many people". Similarly, another participant described monogamy with a main sexual partner as the way in which they prevent HIV and described never using condoms with any partners.

> *Staying with each other [is how my partner and I prevent HIV]. Honestly, I don't use condoms. I've never used condoms. . .I used to be a ho, so I'm not even going to lie. . .I have a pretty face. I can get dudes to do things for me without having to give up anything. And plus too, at the time, I guess this might be important, I didn't want anymore bodies. . .(Black/Female/22 years old)*

Participants also described their existing awareness of PrEP and questions they had related to accessing PrEP along with broader sexual health needs. Some participants described passing familiarity with PrEP, such as "probably heard it on TV or something, or an ad I've seen". No participants said they were very knowledgeable of or had learned in-depth information about PrEP. One participant expressed feeling perturbed that he had never heard about PrEP.

> . . .I had no clue what you guys were talking about at first. I was like, "What's this?" I did not know exactly what it is. It's crazy. I feel like I'm pretty up to date on things but I had no idea. . .(Black/Male/18 years old)

Quantitatively, participants under the age of 18 were most likely to report never having received an HIV test (81%), followed by adults aged 21+ (60%) and 18–20-year-olds (56%). When accounting for sexual activity, 96% of non-sexually active respondents and 48% of sexually active respondents had never been tested for HIV. Previous awareness of PrEP varied such that about half of male and female respondents and all transgender respondents (n = 5) were familiar with PrEP.

*3.7. Multilevel Stigma and Stigma Reduction*

Qualitatively, participants described perceiving multiple dimensions of stigma, some of which were discrimination due to minoritized sexual or gender identities. Several participants described observing stigmatization of participants of LGBTQ+ experience and also described their own personal experiences. For example, one participant described having adults in their life "not respecting my pronouns" and perceiving that they chose not to do so "because they did not want to." Another described being misgendered and avoiding speaking to healthcare providers.

> . . .even some of my family members who have just misgendered me on purpose, or just didn't believe me, like those instances have caused me to be close off health care providers that I know aren't going to support me, or I know who are going to judge me until I look a particular way they think I should look. . .(Black/Transgender/19 years old)

Multiple participants described stigma related to HIV in relation to LGBTQ identity.

> I know for me, I try to expand and not be . . . prejudice about people or all of that stuff. . . I know for my community, I guess one of the stigmas has been, I know even in history too, like the people in LBGTQ community, all of that stuff about HIV or people who are gay, it doesn't make sense to me. . .(Black/Female/17 years old)

One participant living with HIV described being stigmatized by a romantic partner finding out about their status as "they just didn't want to talk to me anymore". The participant cited lack of knowledge as the reason, and that "I wish people was educated about the stigma and there wasn't any stigma on this". Some participants described PrEP as something they personally would not stigmatize because "at least they're trying to prevent getting an infection or that virus".

Participants had recommendations for reducing the stigma of HIV testing. Specific recommendations included more routine testing "from a younger age, and talk about it but don't make fun," along with more comprehensive and consistent sexual health education in schools. Another suggestion was routinizing HIV testing even when participants are not sexually active so they have peer (e.g., accompanying them) and social support (e.g., encouraging them).

> I believe everyone should get tested for having it, even if they haven't had any sex with anybody. . . I believe it'll bring a positive aspect into getting tested. . . [to reduce PrEP stigma]. The best way. I mean, we can at least, I guess, take it with them as showing support that you stand behind them. . . That they have support behind them. They're not the only ones that feel like they're going through it alone. (Black/Female/13 years old)

## 4. Discussion

The present research explored the experiences of Black youth around (1) sexual health service access; (2) HIV familiarity and PrEP awareness; (3) social interactions related to seeking health information; and (4) lived experiences of the early portion of the COVID-19 global pandemic. Our mixed-methods study illuminated multiple findings that were both salient in the qualitative data and supported by the quantitative data. Overall, our sample of Black youth living in the South indicated that in addition to unmet sexual and mental health needs being exacerbated by COVID, these youth experience both stigma and emotional discomfort (e.g., shame, embarrassment) when seeking information about sexual health. These factors likely interfere with learning about sexual health practices broadly. Specific to HIV, HIV risk perception and knowledge/awareness of PrEP were low among cisgender participants. Transgender youth in our sample, however, were far more knowledgeable about HIV risk and PrEP compared to cisgender youth. These types of differences among Black youth may likely stymie efforts to de-stigmatize and normalize sexual health practices among Black youth living in the South and reinforce the need for more communication about HIV/PrEP in Black communities broadly and with different targeted approaches for youth who are not cisgender and may have more specific informational needs given likely familiarity with HIV testing and PrEP.

Youth in our sample were co-experiencing myriad financial and health stressors and those were exacerbated by the pandemic. Unemployment rates were extremely high among adults of working age even before COVID-19, and several participants qualitatively described losing their jobs due to COVID-19 specifically. This finding is confirmed by numerous other studies documenting the impact of COVID-19 disproportionately affecting youth of color in the South and is detrimental to their access to care including sexual healthcare [27–29]. A few interviewees described thinking even less about sexual healthcare due to limited social engagements during lockdown, but disparities in access predated this time period. Additionally, mental health distress was also described by participants and is well supported in the literature as a consequence of the pandemic as well [30]. More research is needed to assess changes in experiences over time, now that pandemic restrictions are largely removed but rates of COVID-19 in Black communities are not comparable to other communities in the United States. Further, these financial hardships invariably minimize the ability of these youth to stay engaged in HIV-preventive care, thereby exacerbating these healthcare disparities.

While many survey participants and interviewees described having access to sexual health services, many did not choose to seek those services regularly. For a minority, it was because they had not had sex yet and therefore saw no reason to interact with providers about this topic. For the majority of others, however, they described feeling discomfort at even broaching the conversation with providers. This was the case for both minors as well as adults, irrespective of having had negative experiences in the past. The main reason described both qualitatively and quantitatively for this avoidance of sexual health services was embarrassment and not specifically due to fear of stigmatization. Previous literature has documented that minors discussing sexual health with providers are often uncomfortable irrespective of whether the provider is affirming and supportive [31]. Therefore, more research is needed to help reduce embarrassment and discomfort in this sexual healthcare encounter, which may include earlier sexual health education resources for minors, more consistent promotion of time alone with providers to discuss these factors, and targeted media and health communication and education targeting parents. Roughly two-thirds of both minors and young adults preferred to speak to either a parent or a healthcare provider about sexual health needs; therefore, targeted communication focused on these audiences and how best to normalize and affirm PrEP use in the broader sexual health conversation is sorely needed. One recommendation for future efforts to address youth discomfort includes providers proactively educating youth about sexual health practices as a segue into inquiring about specific questions youth have. Our findings and previous work suggest that providers have substantial opportunities to cultivate open

discussion and broader conversations of youth's needs than just HIV prevention, which can have large implications for their long-term engagement in sexual healthcare [15,16,32].

Third, our findings show a large disconnect between the actual elevated HIV prevalence amongst Black youth in the South and their awareness of their own lifetime likelihood of being exposed to or diagnosed with HIV. Over half of our sample had not been tested for HIV, and this was unrelated to whether or not they were sexually active. Among cisgender participants, even when there was awareness of PrEP, their confidence in their knowledge of PrEP was relatively low. LGBTQ+ participants reported significant barriers to sexual health services due to barriers like cost, and lack of affirming care from providers who knew how to handle all of their sexual health needs like hormone therapy. This underscores the need for more interventions which (a) address sexual health and gender-affirming needs holistically and not simply HIV prevention, and (b) move towards more interventions that incorporate more meaningful concepts besides perceived HIV risk for youth regardless of their sexual orientation or gender identity. Youth do not accurately assess their HIV risk and find the language of risk stigmatizing [32–34].

This study and its findings are subject to several limitations. First, our data are cross-sectional and cannot assess causality. Second, our study was exploratory and, therefore, only descriptive statistics are presented which cannot be generalized to all needs of Black adolescents in the South. Third, this study did not include an exhaustive list of potential variables to explore, such as substance use and mental health needs. Similarly, our quantitative survey was already completed at the time of the COVID-19 global pandemic; therefore, we only explored its impact on our study population in qualitative interviews. Our results in Table 4 note the social determinant experiences by respondents that were relevant to the experience of the COVID-19 global pandemic in this population (e.g., housing stability, joblessness). Next, while some gender diversity was reflected in the qualitative interview participants, the majority of the survey respondents were cisgender and heterosexual. Therefore, less is known about the needs of Black youth who identify as LGBTQ+ in this sample.

## 5. Conclusions

Our findings contribute greater support for more nuanced and holistic interventions that provide sexual health education to youth as well as their trusted adults and parents, and that address both the stigmatization of HIV and the embarrassment of having candid sexual health conversations. Beyond HIV, these youth are navigating tenuous social determinants like sociodemographic inequities and greater rates of violence at the community level. Patient-centered care must understand and address these needs holistically. Sexual health and PrEP care engagement were not large priorities for these youth, especially given the other issues they navigated—therefore, making access to care simple, routine, free, and affirming rather than stigmatizing is a requisite first step. The importance of culturally affirming work in the context of the US South cannot be overstated for Black youth who identify as LGBTQ+. At the time of this study, copious anti-LGBTQ legislation has been introduced in the region including barring of gender-affirming care by insurance companies. Such policies and institutional barriers compound the challenges these youth are navigating and further add to the stigmatization of the LGBTQ+ community, which has deleterious effects (e.g., suicidal ideation, depression, anxiety, HIV infection, and experiences of violence) [35]. Beyond the social determinants of access to quality healthcare, advocacy at the research, community, policy, and healthcare levels can help to stem some of these harmful policies and ultimately help to reduce some of the health inequities this population continues to experience. Mixed-methods studies that can both quantify and provide an opportunity to provide a platform for youth to discuss their experience in their own words are needed to continue these efforts and mitigate the ongoing disparities this population experiences.

**Author Contributions:** Conceptualization, A.C.M.-B., J.T.M., M.C.D.S., A.L., A.C.; methodology, A.C.M.-B., J.T.M.; software, A.C.M.-B.; validation, M.C.D.S., J.T.M., N.L.B, formal analysis, A.C.M.-B., N.L.B.; investigation, A.C.M.-B., J.T.M., M.C.D.S., N.L.B., L.R.; resources, L.R., A.L., A.C.; data curation, A.C.M.-B., N.L.B.; writing—original draft preparation, A.C.M.-B.; writing—review and editing, M.C.D.S., J.T.M., N.L.B., A.C.; visualization, A.C.M.-B., M.C.D.S., J.T.M.; supervision, A.C.M.-B., J.T.M., M.C.D.S., A.L., A.C.; project administration, A.C.M.-B., J.T.M., M.C.D.S., A.L., A.C.; funding acquisition, A.C.M.-B., J.T.M., M.C.D.S., A.L., A.C. All authors have read and agreed to the published version of the manuscript.

**Funding:** This research was supported by a grant from the Duke-UNC Joint Center for AIDS Research (CFAR) Program (5P30 AI064518 and P30-AI050410-22).

**Institutional Review Board Statement:** The study was conducted in accordance with the Declaration of Helsinki, and approved by the Institutional Review Board (FHI 360, RTI International, Duke University, UNC Chapel Hill; IRB# IGHID 11850).

**Informed Consent Statement:** Informed consent was obtained from all subjects involved in the study Participants were asked to consent to completing screener questions and the survey using verbal consent; those who were interested in being contacted for additional study activities were invited to complete interviews which required written consent.

**Data Availability Statement:** The datasets presented in this article are not readily available because of confidentiality concerns due to the sensitive and identifying nature of qualitative data.

**Conflicts of Interest:** The authors declare no conflicts of interest.

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
