# Peer review of "“I Had No Idea about This:” A Mixed-Methods Exploration of Sexual Health and HIV Prevention Needs among Black Youth in a Southern City"

_adolescents, doi:10.3390/adolescents4020020_

Round 1

Reviewer 1 Report

Comments and Suggestions for Authors

The authors conducted a survey study to assess awareness and attitudes toward PrEP among black adolescents. The samples were convenient samples from a city of NC. The quantitative analysis was based on 83 subjects and qualitative analysis was based on 13 subjects. The demographic distributions from these 13 subjects were different from these 83 subjects. I am having reservation that the results from these convenient samples can be generalized to black adolescents in the South.

Line 78, how was the city chosen?

Line 80, how was the prevalence compared to the other part of the South?

Line 181, it should be Table 1 and Table 2.

Line 182, the total number of age 18 or older is 54 (Table 2).

Reviewer 2 Report

Comments and Suggestions for Authors

Thank you for this interesting paper providing some useful insights into issues around the sexual health intervention needs of Black young people in the southern US. 

The paper is generally well written and structured, though I do have some comments that I hope are helpful in improving its quality. 

One minor comment – a central theme of the study seems to be PreP, and I wonder if this should be included in the title. This is just a suggestion.

Here are my comments on each section. 

INTRODUCTION

This section provides a helpful and detailed background and justifies why the study was undertaken. You provide sufficient detail of the context, and the current situation that clearly requires some attention. 

METHODS

This section provides a detailed overview of your methodology and how you gathered and analysed the data. The mixed methods approach you’ve taken is interesting, especially using the qualitative data to lead the analytical process, drawing on quantitative data to add additional detail (studies often do this the other way around). 

There is a typo in line 78 – this should be ‘quantitative’ survey, I presume? 

The measures section here is useful, though I would recommend moving table 1 to an appendix/end of the paper. 

I note that ethical approval for the study was granted, and that you obtained suitable consent from participants.

RESULTS

These are presented clearly, and I appreciated the detail in table 4 linking qualitative findings with the survey. I would recommend, simply for layout purposes, reworking tables 2 and 3 either into text, or into a more concise table. The current format is a little unwieldly. 

Table 4 is indeed significant, and the quotation excerpts you select are illustrative (though would benefit from some editing). The column ‘quantitative frequencies’ needs some attention, in that it’s not always clear what the text is saying (different types of bullet points etc.). But I can appreciate the points you are making here, especially around HIV knowledge and the intersectional nature of stigma. 

That said, the text on page 13 onwards, which contains useful data, seems to be repeating the essential points included in table 4. My suggestion would be to remove table 4 (even though it is useful data) and use the text section as the basis for reporting your findings – which also includes important aspects of your findings. 

The ‘barriers to seeking sexual healthcare’ is very important, especially around financial and/or social factors that may prevent young people accessing sexual health services. Results in the data around PrEP awareness (or not) are especially intriguing – PrEP is often difficult to scale-up, partly because of the need to encourage people to take a medicine while ‘well’, and the lack of PrEP availability. This data should provide useful guidance for future PrEP programmes.

One additional query – the qualitative data clearly includes content on the impact of COVID-19, though I couldn’t see a direct link with the survey data, which seems to be more generally about employment issues and high rates of joblessness.

 DISCUSSION/CONCLUSION

You draw on your findings, provide a useful summary, and discuss how these fit with similar studies elsewhere. You highlight the significant aspects of your findings and make useful recommendations. 

I also note that you include the limitations of your study, which are appropriate. 

REVIEWER RECOMMENDATIONS 

1.     Consider including PrEP in the title.

2.     Correct the typo on page 2 line 78.

3.     Move table 1 to the end of the paper.

4.     Rework tables 2 and 3 into a more concise table format, or text.

5.     Consider removing table 4 and integrating what is necessary into the text section on page 13 onwards. If you keep table 4, make the ‘quantitative frequencies’ column easier to appreciate, and make the table more concise overall. 

6.     Make more explicit how survey data about employment difficulties links with the COVID-19 epidemic theme (to match the qualitative data, which do indicate this). 

Reviewer 3 Report

Comments and Suggestions for Authors

The manuscript is altogether interesting, revolving around a topic of maximum importance such as the knowledge about sexually transmitted diseases and their prevention among young people and sexual and gender minorities. 

Methods should be revised and exposed in a more readable and understandable fashion. For example: how many questions were there in the survey? In the in-depth interviews all the participants were  asked the same questions?

Line 109: can you clarify what you mean with the sentence "They signed consent forms on participants’ behalf prior to initiation of the interview"? The team members signed consent forms and not the participants?

The COVID part of the survey seems out of place and forced, considering the main focus of the paper is knowledge on sexually transmitted infections and sexual care in general. Since COVID played as a financial and health stressor, it should be mentioned as a contributor in that sense, but the whole section feels out of theme.

Over the aforementioned concerns, the paper is very interesting and well written.

Reviewer 4 Report

Comments and Suggestions for Authors

Thanks for allowing me to review your work. Overall, the manuscript is well crafted and I have only some minor suggestions:

1. Please order the key words alphabetically.

2. Did you have two waves of enrollment based on the quantitative and qualitative part?

3. What qualitative paradigm did the analysis follow? Content analysis? Please provide a reference.

4. In table four what does this mean" Among out quantitative sample"

5. What does this mean: 100% of TG new (but n=5) what are they new to?

4. 

Round 2

Reviewer 3 Report

Comments and Suggestions for Authors

All my previous concerns have been adequately  addressed and the paper has been revised.